# Revisiting Motor Imagery Guidelines in a Tropical Climate: The Time-of-Day Effect

**DOI:** 10.3390/ijerph20105855

**Published:** 2023-05-18

**Authors:** Vanessa Hatchi, Aymeric Guillot, Nicolas Robin

**Affiliations:** 1Laboratory “Adaptation au Climat Tropical, Exercice & Santé” (UPRES EA 3596), Faculté des Sciences du Sport de Pointe-à-Pitre, Campus Fouillole, Université des Antilles, BP 592, CEDEX, 97159 Pointe-à-Pitre, France; 2Inter-University Laboratory of Human Movement Biology-EA 7424, University of Lyon, University Claude Bernard Lyon 1, 69622 Villeurbanne, France

**Keywords:** motor imagery, imagery ability, tropical environment, mental chronometry, time of day

## Abstract

(1) Background: Motor imagery (MI) is relevantly used to improve motor performance and promote rehabilitation. As MI ability and vividness can be affected by circadian modulation, it has been proposed that MI should ideally be performed between 2 p.m. and 8 p.m. Whether such a recommendation remains effective in a hot and humid environment, such as a tropical climate, remains unknown. (2) Methods: A total of 35 acclimatized participants completed a MI questionnaire and a mental chronometry test at 7 a.m., 11 a.m., 2 p.m., and 6 p.m. Visual (VI) and kinesthetic imagery (KI) abilities, as well as temporal congruence between actual walking and MI, were collected. Ambient temperature, chronotypes, thermal comfort, affect, and fatigue were also measured. (3) Results: VI scores were higher at 6 p.m. than at 7 a.m., 11 a.m., and 2 p.m., and temporal congruence was higher at 6 p.m. than at 7 a.m. Comfort, thermal sensation, and positive affect scores were higher at 7 a.m. and 6 p.m. (4) Conclusion: Data support greater imagery ability and accuracy when participants perceive the environment as more pleasant and comfortable. MI guidelines typically provided in neutral climates should therefore be adapted to tropical climates, with MI training sessions ideally scheduled in the late afternoon.

## 1. Introduction

Motor imagery (MI) is a conscious process during which participants internally simulate an action without engaging in its actual execution [1]. While MI is a multisensorial experience, different MI modalities are typically distinguished both in practical imagery interventions and in the scientific literature. MI practitioners first use imagery to mentally see themselves performing an action through two visual modalities: internal visual imagery (IVI, which consists of imagining the action in the first-person perspective or seeing the scene through one’s own eyes) and external visual imagery (EVI, which requires rehearsing the motor sequence from a third-person perspective, i.e., as a spectator like a camera behind oneself) imagery. Kinesthetic imagery (KI) rather involves the sensations of how it feels to perform an action, including the force and effort perceived during movement and balance [2]. 

Over the last three decades, MI has become a very popular technique in the motor field, and a great number of experimental studies have supported that MI contributes to achieve excellence and improve motor performance, increase confidence and intrinsic motivation, manage stress and anxiety, and promote motor recovery [3,4,5,6]. The positive effects of MI may be explained by its functional similarity to actual practice. Accordingly, brain-imaging studies have provided strong evidence that MI activates overlapping patterns of cerebral activation with the corresponding actual execution of the same movement [7,8]. Interestingly, cerebral plasticity occurring after physical practice has also been observed during MI [9,10]. 

Experimental research has further demonstrated, using mental chronometry paradigms [11,12], the temporal congruence between MI and actual movement times [13]. Taken together, these data support that MI might be considered an intermediate form of motor behavior in the continuum extending from the pure mental evocation of a movement to its actual execution and the predispositions to benefit from MI practice. 

From an applied perspective and based on several imagery models and frameworks, there is now considerable evidence of the main influencing variables that are likely to facilitate MI practice and promote its beneficial effects. A large amount of experimental research aimed at determining the optimal MI practice guidelines and provided a comprehensive overview of the main recommendations to develop effective interventions [14]. Specific recommendations, including, for instance, the periodization and dose delivery of imagery sessions, the context of imagery practice, or the content of MI experience, can now easily be provided to athletes to potentiate the benefits of MI training interventions. In addition, the importance of individual MI ability is well documented and should be thoroughly considered when designing imagery training programs [15,16,17]. 

However, the influence of circadian rhythms has received far less attention. In the first study looking at this issue, Gueugneau et al. [18] found that the temporal congruence between MI and actual execution was higher in the early and late afternoon (i.e., between 2 p.m. and 8 p.m.) than in the morning (i.e., between 8 a.m. and 11 a.m.) or late evening (11 p.m.). The authors proposed that these differences could be explained by individual modulations of circadian rhythms throughout the day. Indeed, many cardiovascular, digestive, or thermoregulatory regulatory mechanisms of the human body are controlled by an internal clock also called “circadian rhythms” [19], which, through fluctuation during the day, could impact some “sensorimotor and mental aspects of motor behavior” [11]. This effect of daily modulations was later replicated [11,20,21,22,23,24], thus supporting the importance of this variable of interest. 

Another quite neglected variable of influence is the environmental climate where MI is performed. As extensively shown in physical practice [25,26,27,28], pioneering data have revealed that MI ability might also be influenced by a tropical climate (TC), particularly in hot and humid environments [29]. Both imagery vividness and ease, as well as the ability to achieve temporal congruence, were found to be negatively impacted compared to MI performed in a neutral climate (NC) [30,31]. TC is characterized by high average temperatures (≈27 °C) and hygrometry levels (more than 75% relative humidity in the air) that exceed evapotranspiration capacities and negatively impact motor and cognitive performances [32,33], as well as thermal comfort and perceived fatigue [34]. Over the course of the day, temperatures in a TC can range from 24 °C (±1 °C) in the early morning and evening, to over 31 °C (±2 °C) between 10 a.m. and 3 p.m. [35]. Such variations over the day should thus certainly alter the ability to perform MI appropriately and effectively, perhaps differently from what has been observed in NC, hence supporting the need to investigate the selective circadian modulations of MI in a TC. 

The purpose of this study was to assess the effects of the time of day on MI ability, psychological factors (i.e., thermal sensation and comfort, positive and negative affects), and the capacity to reach temporal congruence when MI is performed in TCs. Based on the negative impact of a TC on cognition and psychological factors mentioned previously, and in contrast to data reported in NC, we hypothesized that the MI abilities of individuals living in TC would be lower at 11 a.m. and 2 p.m., when environmental temperatures are at their highest level, compared to early morning and late afternoon. We further postulated that higher temperatures at 11 a.m. and 2 p.m. would also negatively affect participants’ thermal comfort, affect, and fatigue sensation compared to less warm times of the day (i.e., the morning at 7 a.m. and late afternoon at 6 p.m.).

## 2. Materials and Methods

### 2.1. Participants

Thirty-six healthy acclimatized students (17 males, 19 females; Mean age = 21.5 years, 18–33 years) volunteered to participate in this study and signed a written consent form. The inclusion criterion was living in the West Indies for more than six consecutive months. The experimental protocol obtained approval from the local ethics committee of the University of Antilles, and this study was conducted in accordance with the Declaration of Helsinki. 

This study was conducted in a warm and humid environment (i.e., a TC) in a rectangular room (20 m × 7 m) where doors and windows were open. Temperatures were measured with a Wet Bulb Globe Temperature sensor (QUESTemp 32 Portable Monitor, QUEST Technologies, Oconomowoc, WI, USA) and ranged from 24 °C to 30 °C. The average relative humidity was 75% (RH, ±10%) throughout the day (Table 1).

### 2.2. Measurement Instruments

An online pre-experimental questionnaire was completed to collect personal information from each participant, including gender, age, health status, physical activity level, approximate menstrual periods for women [36,37], and the number of months spent in a tropical environment [34]. In addition, the chronotype of the participants was assessed using the Morningness–Eveningness questionnaire [38]. This questionnaire consists of a subjective assessment of circadian typologies using 19 multiple-choice questions on waking, bedtime, and situation preferences. A score based on morning–evening preference was calculated and determined participants as “totally morning” (score > to 70), “moderately morning” (59 < score < 69), “neutral or intermediate” (42 < score < 58), “moderately evening” (31 < score < 41), or “totally evening” (score < 30) [39]. 

Participants’ sleep quality was also assessed using the Pittsburgh Sleep Quality Index (PSQI) [40]. This questionnaire consists of 19 items assessing subjective sleep quality, rest time, sleep disturbance, habitual sleep efficiency, use of sleeping pills, and daytime dysfunction in the month prior to the study. This test was administered to check for the absence of obvious disturbances in sleep/wake cycles and to check for predisposition to benefit from the natural effects of sleep [41]. 

Then, at each experimental session scheduled at 7 a.m., 11 a.m., 2 p.m., and 6 p.m., several dependent variables were measured: participants’ feelings of fatigue, comfort, and thermal sensation were measured using self-reported Likert scales ranging from 1 (“not tired at all”) to 7 (“totally tired”), from −3 (“very uncomfortable”) to 3 (“very comfortable”), and from −3 (“very cold”) to 3 (“very hot”) [41], respectively.

Affect was assessed with the Positive Affect and Negative Affect Schedule (PANAS) [42]. This questionnaire evaluates emotional states on a Likert scale ranging from 1 (“not at all”) to 5 (“extremely”). It comprises 20 items, including 10 positive affects (active, alert, attentive, determined, enthusiastic, excited, inspired, interested, proud, and strong) and 10 negative affects (scared, ashamed, anxious, guilty, hostile, irritable, nervous, frightened, and upset).

MI ability was assessed using the MIQ3-f [15], which consists of 4 simple arm and leg movements. After physically performing each movement, participants were asked to imagine themselves performing the same action in a predetermined imagery type (IVI, EVI, or KI). For each trial, the participants were asked to rate the ease of their imagery using Likert scales ranging from 1 (i.e., “very difficult to see/feel”) to 7 (i.e., “very easy to see/feel”). A higher score represented a greater ease of imaging. 

Finally, the mental timing test consisted of measuring actual and imagined walking times. Participants were asked to walk or imagine walking at a freely chosen speed towards a target located 10 m away. The durations of the 20 trials (10 real and 10 randomized imagined trials) were measured by the participants using a stopwatch [31]. No information regarding the actual or imagined walking durations was provided.

### 2.3. Experimental Procedure

The experimental phase consisted of four sessions (repeated measures) spread over a single day. During each session, participants were asked to answer questions concerning fatigue, comfort, and thermal sensations, to complete the PANAS questionnaire, perform the MIQ-3f, and perform the mental chronometry test. To avoid any order effect, these measures were collected in a random order throughout the experiment. 

Sessions were conducted in the early morning (7 a.m.), late morning (11 a.m.), early afternoon (2 p.m.), and late afternoon (6 p.m.). 

### 2.4. Data Analysis

Data from one participant were excluded because the PSQI score revealed poor sleep quality (score = 14). Statistical analyses were performed using “STATISTICA” software (12.0, StatSoft, Paris, France) based on 35 participants (17 males and 18 females; Mean age = 21.6 years, 18–33 years). 

For each MI modality (i.e., IVI, EVI, and KI), the MIQ-3f scores were taken into account. For the mental chronometry test, the isochrony index was obtained by calculating the absolute difference between average actual and imagined walking times [11]. The closer the isochrony index was to zero, the better the performance was. For these two tests, an analysis of variance (ANOVA) was performed on four different time slots (7 a.m., 11 a.m., 2 p.m., and 6 p.m.). These were used as independent variables. 

Repeated measure ANOVAs were then performed with the same experimental design for comfort and thermal sensation scores, positive affect, negative affect, and fatigue sensation. For each variable of interest, the homogeneity of variances (Levene’s test) was checked, and Kolmogorov–Smirnoff tests revealed that data were normally distributed. Post-hoc analyses were performed using Newman–Keuls tests, and the alpha threshold for the type 1 error rate was set at 5%.

## 3. Results

The results regarding the chronotypes of the participants, evaluated using the Morningness–Eveningness questionnaire of Horne and Ostberg [38], are illustrated in Table 2.

### 3.1. MIQ Test Scores

As the ANOVAs for all dependent variables revealed no main effect of gender, nor any interaction between gender and time of day (all *p*s > 0.05), only the results for the main effects of time of day are presented below.

The ANOVA performed on IVI scores revealed a significant time-of-day effect (F(3, 105) = 4.39, *p* = 0.006, ηp2 = 0.12, Figure 1a). Post-hoc analysis revealed that IVI scores were higher in the late afternoon (6 p.m.) than in the other time slots (*p*s < 0.05).

EVI scores showed a significant effect of time of day (F(3, 105) = 5.80, *p* = 0.001, and ηp2 = 0.15, Figure 1b), being lower in the early morning (7 a.m.) than in other time slots (*p*s < 0.05), and that scores at 6 p.m. tended to be higher than those obtained at 11 a.m. and 2 p.m. (*p* = 0.098 and *p* = 0.061, respectively). However, there was no significant time-of-day effect on KI scores (F(3, 105) = 0.76, *p* = 0.521, and ηp2 = 0.02).

### 3.2. Mental Chronometry Test

Similarly, a significant effect of time of day on the isochrony index (F(3, 105) = 4.37, *p* = 0.006, ηp2 = 0.11, Figure 2) showed a higher mental chronometry performance (i.e., lower isochrony index, closer to 0) in the early afternoon (2 p.m.) and late afternoon (6 p.m.) than in the early morning (7 a.m., *p*s < 0.05).

### 3.3. Psychological Factors Scores

For thermal comfort, the ANOVA revealed a time-of-day effect (F(3, 105) = 4.97, *p* = 0.003, and ηp2 = 0.12, Figure 3a) with higher scores at 7 a.m. and 6 p.m. than at 11 a.m. and 2 p.m. (*p*s < 0.05). Similar but larger effects of time of day on thermal sensations (F(3, 105) = 17.81, *p* = 0.000, and ηp2 = 0.34, Figure 3b) showed higher scores at 11 a.m. and 2 p.m. than at 7 a.m. and 6 p.m. (*p*s < 0.001).

For perceived fatigue, a time-of-day effect was observed (F(3, 102) = 3.98, *p* = 0.017, and ηp2 = 0.1, Figure 3c), scores being higher at 2 p.m. than at 6 p.m. (*p*s < 0.05).

A significant time-of-day effect on positive affect was observed (F(3, 105) = 5.89, *p* = 0.001, and ηp2 = 0.14, Figure 4a). Scores were higher at 7 a.m. and 6 p.m. than at 11 a.m. and 2 p.m. (*p*s < 0.05). Similarly, the data also showed a significant time-of-day effect on negative affect (F(3, 105) = 3.67, *p* = 0.015, and ηp2 = 0.09, Figure 4b), with higher scores at 7 a.m. than at 6 p.m. (*p*s < 0.05).

Finally, data revealed no time-of-day effect on participants’ motivation (F(3, 105) = 1.93, *p* = 0.130, ηp2 = 0.05).

## 4. Discussion

The primary objective of this study was to assess the time-of-day effects on MI abilities of young acclimatized adults in a TC in order to update and adjust the usual MI guidelines and recommendations in extreme environments [14]. 

We first postulated that MI ability would be negatively impacted at 11 a.m. and 2 p.m., due to environmental constraints (i.e., when temperatures are at their highest) compared to early morning (7 a.m.) and late afternoon (6 p.m.). Data supported this hypothesis as participants reached higher IVI and EVI scores and greater ability to achieve temporal congruence at 6 p.m. than at 11 a.m. and 2 p.m. As the respective mean temperatures were 26.4 °C, 30.2 °C, and 30.9 °C, data therefore confirmed the negative effect of heat stress on cognitive performance [33,43]. 

Robin et al. [31] showed that the temporal congruence between actual and imagined walking times was lower in a TC (30 °C) than in a NC (24 °C), hence postulating that thermal stress was likely to negatively affect cognitive performance. However, they did not consider the individual perception of the environment (e.g., comfort and thermal sensations) nor related psychological factors (e.g., affect and perceived fatigue), which were expected to be affected by a TC as well and to further negatively impact cognitive performance [44]. In a second separate study, Robin et al. [34] showed that TCs contributed to a decrease in positive affect and thermal comfort scores and further increased fatigue and thermal sensation scores. They argued that it was certainly more pleasant for participants to be in a NC than in a TC before engaging in a MI task. Present data combining the perception of the environment and MI corroborate these findings, and showed that positive affect and thermal comfort were lower when environmental temperatures were high, and that thermal and fatigue sensations were higher when environmental temperatures were low. Taken together, the participants perceived the environment to be more comfortable in the late afternoon (6 p.m.) than in both the late morning (11 a.m.) and early afternoon (2 p.m.).

One of the particularities of this research was to combine several measures of dependent variables, which made it possible to postulate, spurred by these findings, that IVI and EVI scores, as well as the ability to achieve temporal congruence, were directly negatively impacted by the environmental temperatures at 11 a.m. and 2 p.m., which induced a decrease in positive affect and thermal comfort, along with an increase in perceived fatigue and thermal sensations. Therefore, scheduling MI in a TC requires adjusting the imagery recommendations which are usually provided in a NC and which consider the influence of circadian rhythms [11,18], in order to fit with specific environmental constraints. As suggested by the Global Worskspace Theory [45], MI and accommodation to high temperatures might compete and share available resources [33], and thermal discomfort, fatigue, and decreased positive affect may still mobilize additional resources and result in exceeding the total capacity of the workspace. In a TC, we advocate that MI should thus be primarily performed in the late afternoon. 

Surprisingly, the data showed that the MI capacity scores and the ability to achieve temporal congruence were lower at 7 a.m. than at 6 p.m., while there was no difference when comparing positive affect, comfort, thermal sensation, fatigue sensation, and motivation scores. This finding revealed different MI abilities and accuracies, although the environmental conditions were perceived as being similar by the participants. It is important to remember that this study was carried out in a TC close to the equator where the sun rises very early (from 5:30 a.m. to 7 a.m.) when university classes and work begin in most companies and administrations. In addition, the studied public has an average wake-up habit of 7:55 a.m., which could explain the low performance at 7:00 a.m. compared to the other studies. Therefore, a first explanation might come from higher self-reported negative affect scores (e.g., participants felt more hostile, irritated, or nervous), despite a perceived pleasant environment. It is possible that the early morning time chosen in this study to perform the first MI trials, which corresponded to the opening hours of school, administrative, and medical institutions usually observed in a TC, was therefore felt to be “too early” by the participants, hence resulting in increased negative affect scores. 

A second explanation could be related to participants’ chronotypes. Indeed, cognitive performances of late chronotypes (evening) have been shown to be significantly impaired compared to early chronotypes (morning) when performed early in the morning [46]. However, in this current study, the majority of participants had “neutral” (*n* = 19) or “moderate evening” (*n* = 6) chronotypes, and none of them fell in the “totally evening” nor the “totally morning” categories. Adan et al. [19] argued that the difference between “morning” and “evening” types lies primarily in physiological variables, so that “morning” type individuals exhibit an earlier circadian phase of melatonin compared to “evening” types and thus wake up earlier. Cox and Olatunji [47] also reported a differential effect of sleep loss on positive and negative affect and suggested that the negative affect of the “evening” type chronotype might increase after sleep loss, potentially explaining the low early morning performance (7 a.m.) and resulting in high negative affect. Given the absence of participants with “totally morning” chronotypes, we cannot exclude the potential influence of chronotype on MI ability performance. More research examining this issue is needed, including a larger sample of participants to test this hypothesis.

It is also important to note that no time-of-day effect was observed on the KI scores. These results corroborate data by Robin et al. [35], who observed, in good imagers, a lack of significant differences between KI scores obtained in a TC (31 °C) and those measured in a NC (24 °C). Finally, data did not reveal any effect of time of day on participants’ motivation. Given that the participants were fully acclimatized to the TC and had lived in that environment for several years, they certainly developed relevant and appropriate psychological adaptations that allowed them to better tolerate the heat and resist motivational decline [33]. It is likely that participants not acclimatized to such hot and humid environments may be more impacted and would experience a decline in motivation at the hottest times of the day. 

As generally found in experimental research work, the present study has some limitations that should be considered before drawing firm conclusions. Firstly, although sleep quality was measured using the PSQI, it is possible that the amount of sleep acquired the night before the experiment (or even before each time slot) could be an interesting data point to further justify a rational or non-rational state of fatigue. Although obtaining “totally-morning” or “totally-evening” participants is quite complex [46,48], a larger sample of participants should be recruited, not only to collect more data but also to study participants from each chronotype category. Secondly, some studies on sprint cycling tasks have shown that body temperatures in tropical environments usually vary over the course of the day [49]. However, research has shown that an increase in body temperature can have an impact, under certain conditions (e.g., very high outdoor temperature and intense physical exercise), on the performance of cognitive tasks [50]. Therefore, it would be interesting to measure and monitor body temperature to properly assess physiological variations and better understand the effects of TCs. Finally, as the participants carried out several sessions during the same day, it is possible that they could have benefited from a learning effect. Further research comparing MI ability, over a longer period (i.e., 4 days to 4 weeks), at different times of the day, and with a larger sample size should be conducted.

## 5. Conclusions

The purpose of this study was to evaluate the effects of the time of day on MI ability and the capacity to achieve temporal congruence, as well as psychological factors (thermal comfort, thermal sensation, fatigue sensation, and affect), when MI is performed in a TC. The data provided clear evidence that visual MI scores and the ability to imagine in real time were better at 6 p.m. than at 7 a.m. The poorest performance observed at 11 a.m. and 2 p.m. can be explained by thermal discomfort that resulted in lower positive affect and higher fatigue. Based on these findings, we suggest updating and adjusting the MI guidelines that are usually provided in NC and to schedule MI sessions in TC at the end of the afternoon for sports, school, and rehabilitation settings (Figure 5).

We propose to target the work primarily around 6 p.m., although the ability of MI to work in later time slots has not yet been tested. Therefore, it is likely that this range extends, as shown by Gueugneau et al. [11], to at least 8 p.m. The extent of this time window would therefore need to be controlled; however, in any case, 6 p.m. is the slot from which work is recommended.

## Figures and Tables

**Figure 1 ijerph-20-05855-f001:**
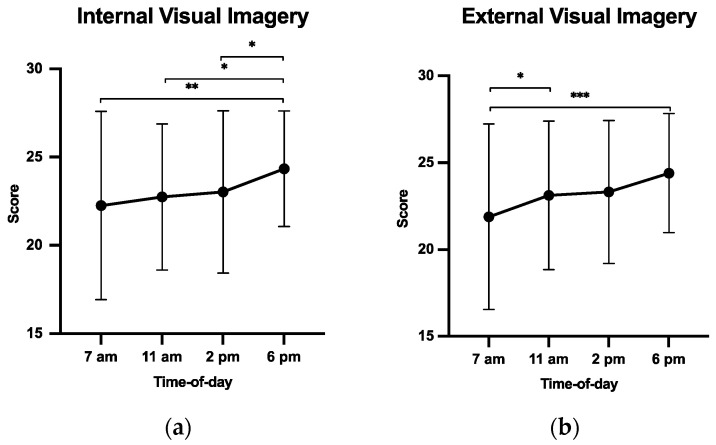
MIQ-3f test scores: IVI (**a**); EVI (**b**) as a function of time of day (h). * *p* < 0.05; ** *p* < 0.01; *** *p* < 0.001.

**Figure 2 ijerph-20-05855-f002:**
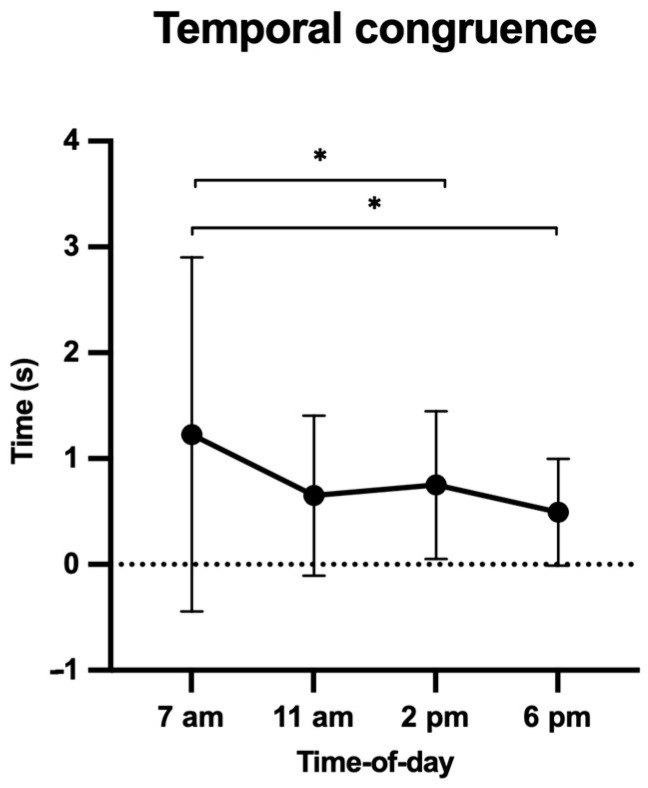
Isochrony index (s) as a function of time of day (h). * *p* < 0.05.

**Figure 3 ijerph-20-05855-f003:**
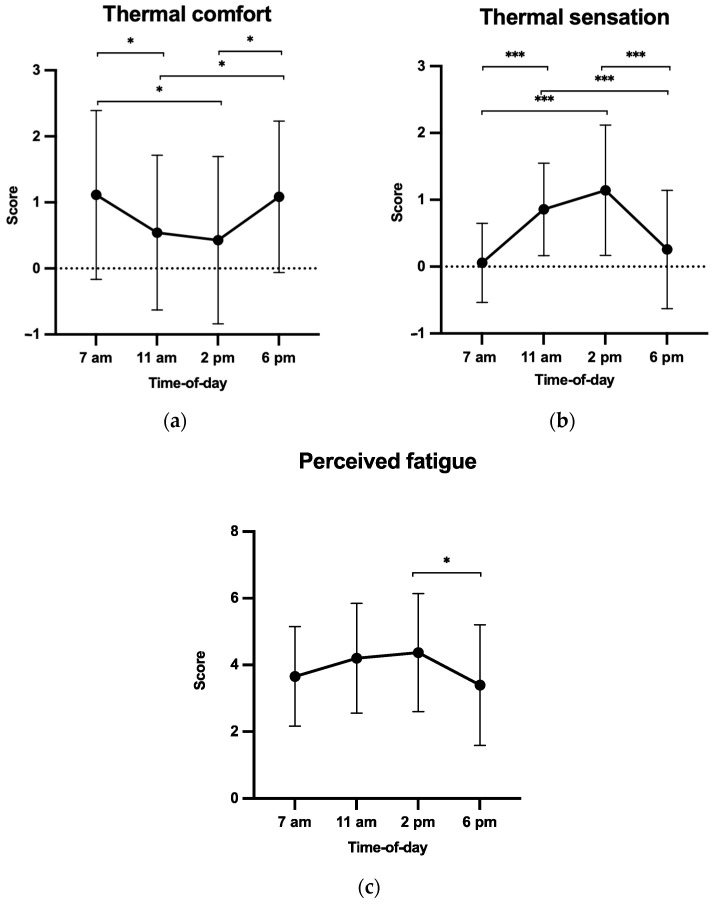
Scores for thermal comfort (**a**), thermal sensation (**b**), and perceived fatigue (**c**) as a function of time of day (h). * *p* < 0.05; *** *p* < 0.001.

**Figure 4 ijerph-20-05855-f004:**
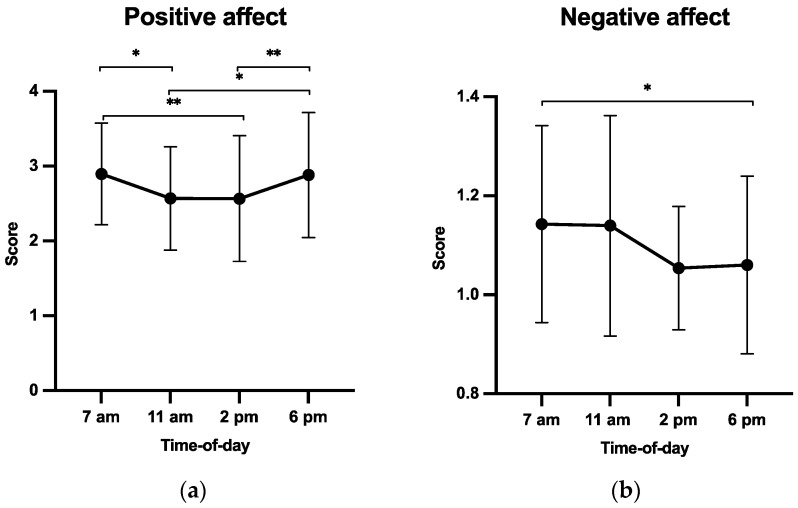
Positive affect (**a**) and negative affect (**b**) scores as a function of time of day (h). * *p* < 0.05; ** *p* < 0.01.

**Figure 5 ijerph-20-05855-f005:**
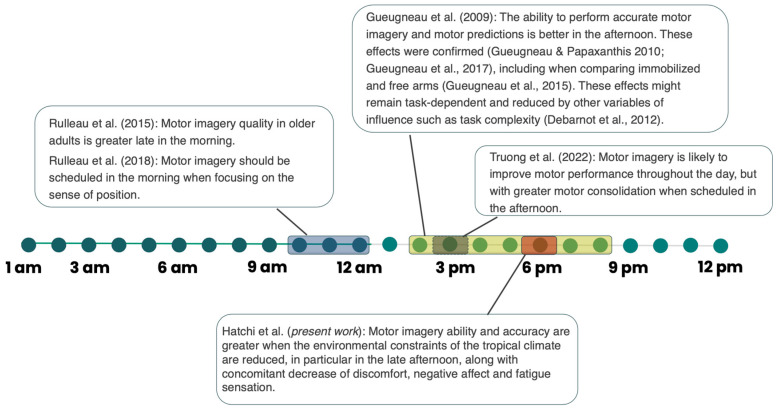
Synthetic view of the time-of-day effect on motor imagery ability and accuracy complementing current recommendations [11,21,22,23,24,51].

**Table 1 ijerph-20-05855-t001:** Temperatures in degrees Celsius according to the time of day. TS: dry temperature; TH: wet temperature; TR: radiant temperature; WBGT: Wet Bulb Globe Temperature.

	TS	TH	TR	WBGT
7 a.m.	21.8	24.7	24.8	22.7
11 a.m.	23.1	30.2	29.5	25.2
2 p.m.	24.3	30.9	30.5	26.3
6 p.m.	23.1	27.7	27.8	24.5

**Table 2 ijerph-20-05855-t002:** Chronotypes of the participants.

Chronotype	Total
Totally morning	0
Moderately morning	10
Neutral	19
Moderately evening	6
Totally evening	0

## Data Availability

All the data used in this study are available. A link will be sent on request.

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
