# Peer review of "Revisiting Motor Imagery Guidelines in a Tropical Climate: The Time-of-Day Effect"

_ijerph, 2023, doi:10.3390/ijerph20105855_

Round 1

Reviewer 1 Report

The article deals with the interesting topic of chronometry of imagery and imagery ability in tropical climates depending on the time of day. The article is well written, but I would recommend dividing the article into smaller paragraphs to better follow the content.

However, I wonder if there may have been a learning effect (participants learned how to perform the test) in the study, so I recommend adding this to the limitation section. Maybe in the future it would be better to compare imagery ability over a longer period of time for example 3-4 weeks at different times of the day. Another option would be to conduct cross-sectional research with a larger number of respondents. I would also recommend to add this information. 

Please correct the first statement in the Background: "Motor imagery (MI) is relevantly used to improve motor performance 12 and promote rehabilitati on" -rehabilitation instead of  rehabilitati on.

 Thank you for your research.

Reviewer 2 Report

Overall, the article is well-written and provides a valuable contribution to the field. However, some improvements could be made to enhance the clarity and focus of the work.

1. The result that the ability to imagine in real-time were better at evening vs morning seems to contradict with many other studies which shows enhanced cognitive ability and performance in the morning rather than evening. Could it be because 7 am is almost mid morning. Early morning could be even earlier than 7 am specially in the tropical climate provided the fact that in places with tropical climate sun rises quite early.

2. Why is the study conducted only on a single day? Will there be any difference in the outcome if it were conducted over a period of time?

3. The method of IVI and EVI is not very clear.

4. Why is it positive and negative "affect" and not "effect"?

Reviewer 3 Report

Review of the manuscript entitled: Revisiting motor imagery guidelines in a tropical climate: the  2 time-of-day effect.

The manuscript submitted is appropriate to the subject matter and scientific rigor. The authors raised a very current issue at work, which is not only interesting from a scientific but also a practical point of view. Some remarks improving the quality of future research. and suggested changes and comments to the submitted manuscript in order to improve the quality of the planned research and future publications below:

1.     Would you please add the DOI or other electronic access in few references, page numbers and correct links as some are incorrect - see comments in attachment.

2.     Line 163: before excluding a person from the study, the authors write "17 males, 19 females; Mage = 21.8 years, 18-33" after excluding one of the women, they write "(17 males and 18 females; Mage = 21.8 years, 18-33 years). I have a question, in the group of women, after excluding one of them, the age result in this group did not change?

3.     Line 281: I think another reference should be given instead of [49], e.g. Global workspace theory of consciousness: toward a cognitive euroscience of human experience, Bernard J Baars; DOI: 10.1016/S0079-6123(05)50004-9

4.     Line 490: Would You please delete items [51] in references - this item is not in the content of the article
